# Enabling Artificial Intelligent Virtual Sensors in an IoT Environment

**DOI:** 10.3390/s23031328

**Published:** 2023-01-24

**Authors:** Georgios Stavropoulos, John Violos, Stylianos Tsanakas, Aris Leivadeas

**Affiliations:** 1Department of Informatics and Telematics, Harokopio University of Athens, 17778 Tavros, Greece; 2Department of Software and IT Engineering, École de Technologie Supérieure, Montreal, QC H3C 1K3, Canada; 3School of Electrical and Computer Engineering, National Technical University of Athens, 15780 Athens, Greece

**Keywords:** virtual sensors, machine learning, regression, Internet of Things, IoT platform, smart homes, Raspberry Pi

## Abstract

The demands for a large number of sensors increase as the proliferation of Internet of Things (IoT) and smart cities applications are continuing at a rapid pace. This also increases the cost of the infrastructure and the installation and maintenance overhead and creates significant performance degradation in the end-to-end communication, monitoring, and orchestration of the various connected devices. In order to solve the problem of increasing sensor demands, this paper suggests replacing physical sensors with machine learning (ML) models. These software-based artificial intelligence models are called virtual sensors. Extensive research and simulation comparisons between fourteen ML models provide a solid ground decision when it comes to the selection of the most accurate model to replace physical sensors, such as temperature and humidity sensors. In this problem at hand, the virtual and physical sensors are designed to be scattered in a smart home, while being connected and run on the same IoT platform. Thus, this paper also introduces a custom lightweight IoT platform that runs on a Raspberry Pi equipped with physical temperature and humidity sensors, which may also execute the virtual sensors. The evaluation results of the devised virtual sensors in a smart home scenario are promising and corroborate the applicability of the proposed methodology.

## 1. Introduction

The Internet of Things (IoT) has allowed for the digitization of almost every aspect of our life by refurbishing applications, introducing new and smarter ones, and stressing the capabilities of the current Internet to its full potential. This revolution of software engineering and information technology creates a lucrative environment to host new ideas for products and services that benefit and improve the productivity of mankind. Specifically, the IoT allows for gathering, storing, communicating, and processing information anywhere and anytime [1]. To do so, IoT devices can extract knowledge by monitoring their environment, making decisions, and automatically operate various types of services or even machines.

The typical architecture of IoT consists of four layers: device layer, network layer, platform layer, and service layer [2]. As shown in Figure 1, various types of IoT devices can be found at the device layer, ranging from simple sensors, such as temperature sensors, to fully operational machines equipped with multiple sensors, such as heating, ventilation, and air conditioning (HVAC). All these devices are generating data that have to be transferred to other IoT devices, applications hosted in the Cloud, or even to be consumed directly by the end-users. Thus, the IoT is based on the integration of network interfaces into devices [3], in which they will establish the necessary communication with the rest of the Internet. Additionally, appropriate middlewares [4,5] can create a management platform for the orchestration of IoT devices and applications. Finally, the data generated by the IoT devices do not remain raw, but they are processed at a service layer [6], in order to allow the applications to make smarter decisions and control smart environments.

At the bottom of this architecture, the very first step to create an IoT system is to select the IoT devices. These devices are so far considered as real devices, usually widely dispersed, with low storage and processing capacity, aiming to improve the reliability, performance, and quality of services in a multitude of applications. Obviously, the number and capabilities of the devices depends on the type of application or industry that the IoT services will be offered. For example, in smart city applications [7], such as transportation, energy, and entertainment, there are significant IoT infrastructure and application requirements. These requirements are proportional to the size of the city population or the number of buildings. This is a phenomenon which is also observed in smaller scale applications, such as smart home [8] use cases, where the infrastructure requirements can be proportional to the number of rooms. This happens because smart homes are controlled and monitored based on the continuously generated data by the sensors placed in every room.

However, there are some serious challenges that need to be addressed, so that the full potential of IoT can be understood. For instance, the increasing number of IoT devices and the necessity of processing a significant amount of data have the disadvantage of increasing (i) the number of transmitted packets [9], (ii) the resource requirements, and (iii) the total overhead of device deployment and synchronization [10]. This growing trend in the demand of IoT devices imposes limitations, regardless of the computing layer it runs, i.e., Cloud, Edge, or Fog [11]. This set the motivation to conduct research on how to offer the same functionalities in a smart home application, while minimizing the number of IoT devices. Hence, this paper proposes the emulation of some physical sensors with ML models. The ultimate goal of the proposed methodology is to make these new virtual and ML-enabled counterparts output values close to the real ones of the replaced physical sensors.

To achieve this goal, multiple experiments are conducted with time series measurements of temperature and humidity in a smart home, and it is observed that, even though the values of the sensors placed in separate rooms are different, they are also related. Thus, a set of different ML models, such as linear, bootstrap aggregating [12], boosting [13], and artificial neural networks (ANNs) [14] are trained based on historical data and concluded with the most accurate model for every room. To maximize the cost efficiency of the proposed solution, only one physical sensor is kept, while the rest are replaced by the proposed ML models. These software models work as virtual sensors that can run on the same IoT device as the physical sensor. The achievement of this methodology is that one physical sensor monitors the data of one room and provides the temperature/humidity of all the rooms in a smart home using the ML models, while keeping a high accuracy.

In order to examine the applicability of the proposed methodology, a platform has been developed that monitors the temperature and humidity in a central room with a DHT11 sensor connected to a Raspberry Pi (RPi). This platform visualizes the time series metrics of one physical and five virtual sensors via a web graphical user interface. The contributions of this paper can be summarized as follows:The use of virtual sensors is proposed, instead of physical ones, in order to decrease the number of IoT devices in a smart home use case.A concrete discussion is provided, regarding the various technical decisions in the design of virtual sensors in an IoT and smart home environment.An extensive experimental comparison among different ML models has been conducted for the development of temperature and humidity virtual sensors.A brief guideline is provided of a basic IoT platform that integrates physical devices, a socket API, a graphical user interface, data storage, and artificial intelligence virtual sensors.

The rest of the paper is structured as follows: Section 2 highlights the related work in IoT, smart homes, sensors, and the use of ML models. Section 3 presents the proposed IoT platform with physical and virtual sensors. Section 4 discusses the utilization of ML models for the development of virtual sensors. Section 5 presents the experimental outcomes and the evaluation results. Section 6 concludes the paper, reports the limits of our work, and suggests future directions.

## 2. Related Work

The related work of this particular research extends over six main research domains. Firstly, the impacts of virtual sensors in various scientific domains are discussed. Secondly, the current status in the development of IoT platforms is presented. Thirdly, it is explained how the contemporary smart homes are equipped with IoT devices and controlled by IoT platforms. Fourth, a presentation of physical sensors and virtual sensors connected with or run on IoT devices is made. Fifth, a brief review of the ML models suitable to develop virtual sensors is given. Towards the end of the section, it is explained how the proposed model fills the research gaps of the related work. Finally, the impacts and other research gaps of virtual sensors are presented.

### 2.1. Impacts of Virtual Sensors

Virtual sensors have applications and multiple impacts in domains such as health-care, entertainment, fitness, sport, digital twins, and Industry 4.0. The first impact to mention is that they extend the reliability and availability in applications, without adding extra hardware complexity and maintenance [15]. They can be the basis for a fault detection mechanism recognizing discrepancies between the sensor data and expected measurements [16]. They provide an alternative when a physical sensor cannot be placed in the preferred position, due to spatial conditions, such as a lack of space or a hostile environment, such as exposure to acids and extreme temperatures [17]. Virtual sensors can replace physical sensors and deliver a higher level of information based on multiple heterogeneous sensor signals [18]. Virtual sensors also have the impact that they overcome a number of weaknesses of physical sensors, such as reducing signal noise [19] and drifts [20]. Drifts constitute a well-known phenomenon rendering a physical sensor inaccurate over time, due to wear or pollution. The last impact to mention is that virtual sensors are extremely flexible and can be redesigned as required, while physical sensors, once installed, often can only be repositioned by mechanical intervention [21].

### 2.2. Internet of Things Platforms

As described in Section 1, an IoT ecosystem consists of four layers, namely the device, network, platform, and service. In this part, more details for the IoT platform layer are given, also referred to as IoT middleware. In particular, this layer is a composition of software components that enable the communication of IoT devices and smart appliances with three purposes [4]: First, to facilitate the acquisition, processing, transformation, organization, and storing of sensor data. Second, to deliver situational awareness through warnings and a graphical user interface. Finally, to take optimal decisions through the control of appliances and actuators.

Many articles discuss and provide reviews regarding the IoT platforms. Zdravkovic et al. [4] presented seventeen IoT platforms and introduced a categorization of them into domain-specific, technology-specific platforms for data acquisition and analysis, full-scale generic IoT platforms, and platforms designed to offer specific services. Agarwal et al. [5] reviewed Cloud-based IoT platforms, such as the AWS IoT Platform, Microsoft Azure IoT Hub, and Google IoT Platform. They provided specific selection criteria, such as availability, deployment type, pricing model, support for required hardware, security, type of communication protocol, storage technologies, and data analytic methods. Nardis et al. [22] brought up the question of whether it is better to use an existing IoT platform or develop a new platform from scratch. They concluded that it is a better choice to develop a new IoT platform, when the goal is to introduce and test new functionalities not available in existing platforms. At this point comes one of the contributions of this paper. Specifically, a guideline is presented for new researchers describing the basic components to develop an IoT platform that is compatible with contemporary technologies, extensible, and works along with the virtual sensors that are also proposed.

### 2.3. Smart Homes

The proposed virtual sensor methodology has been designed and tested in a smart home use case. Smart homes are inherently related to IoT and sensors’ technologies, since they use sensory data to assess the current state of the home, predict the inhabitants’ intent, and act preemptively based on the intent assumptions [23]. Hence, the integration of smart devices with sensors, actuators, and decision-making units optimize the comfort, safety, and wellness of the inhabitants, adapting the home to their behavior and providing ambient intelligence.

The data generated by the home sensors are the basis for context awareness that provides building-focused event automation and user-focused remote monitoring and control [8]. Smart appliances, IoT devices, and specifically sensors work continuously and generate massive amounts of data. These data should be transferred to Cloud infrastructures or should be processed and stored locally at the smart home for optimal decision making and keeping historical records [7]. Both computing and storing approaches incur overheads that come from the number of IoT devices, the volume of the generated data, the speed at which data are generated, stored, and processed, and the heterogeneity in data structures [24]. A continuous transfer of large amounts of IoT data and from multiple devices to the cloud burdens the network’s bandwidth and incurs privacy and security issues. Recent studies show that edge computing is more suitable for IoT than other computing paradigms [9]. Nevertheless, the edge incurs storage and processing limitations. In this research, ways to overcome these resource limitations are examined, not by increasing the computing nodes, but by decreasing the generated data. It has been observed that, very often, the sensing metrics, although they are different from one from the other, are not independent. This observation led to the use of the values of one sensor and inferred the values of the other sensors. In this way, the amount of generated data can be significantly decreased, without sacrificing the smart home context awareness. This approach is different from the existing works that try to achieve home context awareness by using prediction ML algorithms for the activities of daily living, as in [6]. In contrast, in this paper, ML models are used to implement virtual sensors.

### 2.4. Physical Sensors and Virtual Sensors

Brunello et al. define a sensor as a device which measures a physical quantity and transforms it into sensor data that can be interpreted by an instrument or an observer [25]. Next, they distinguish the physical sensors, which measure physical phenomena directly from the virtual sensors that are software-based models. The virtual sensors produce sensor data by fusing data received synchronously or asynchronously from physical or other virtual sensors [18]. Comparing physical to virtual sensors shows that physical sensors may be prone to noise, interfere with each other, lose accuracy over time, or their use is even technically not feasible, due to spatial or environmental conditions. On the other hand, virtual sensors replace a subset of physical sensors with virtual ones, allowing for the monitoring of unreachable locations, reducing the sensors deployment costs, providing a fallback solution, and finally, improving the reliability of physical systems [15].

There are three main types of approaches to designing a virtual sensor model: (a) mechanism-based, (b) knowledge-based, and (c) data-driven. What distinguishes these three types of approaches is how the relation between the input parameters and the output sensor value is defined. The relations can be based on physical laws, expert knowledge, or a model that recognizes data patterns. Specifically, the mechanism-based [26] virtual sensors are constructed based on the behavior of the operating mechanisms to be measured, describing the essential physical correspondence between the input and output quantities. The knowledge-based sensors [27] use the knowledge of experts who explicitly define the equations or rules between the input values and the sensor output. The data-driven [28] methods have become the mainstream approach for training AI mechanisms based on historical data. Most data-driven virtual sensors apply a data fusion technique to provide precise measurements of one specific phenomenon or an abstract representation of diverse sensor inputs [29]. This paper’s proposed virtual sensor methodology, instead of aggregating the inputs of physical sensors, replaces every single physical sensor with a virtual one.

Brunello et al.’s work [25] is the closest work to our paper. They also use virtual sensors based on ML models to estimate the temperatures in indoor environments. The main points in which the paper’s proposed methodology differs from Brunello et al.’s work are the following:Brunello et al. proposed an automatic selection technique to find the minimum number of physical sensors and their optimal positions in an indoor space, while the new proposed model uses only one physical sensor that can be placed in any spot in a smart home.Brunello et al. input the measurements of the physical sensors into one global data-driven model that provides virtual measurements, while the new proposed model uses a separate data-driven model for every virtual sensor.Brunello et al.’s virtual sensors estimate only temperatures, while the new proposed jointly estimates the temperature and humidity.Brunello et al. presented only the design of the virtual sensor, while the new proposed methodology also takes into consideration the software platform and the processing device on which the physical sensors are connected and the virtual sensors run.

### 2.5. Machine Learning

Virtual sensors based on data-driven models have been used for more than a decade. Kaedlec et al. reviewed virtual sensors in a smart industry context [28]. They presented the characteristics that ensure the accuracy and quality of sensor data including: (a) missing values, (b) data outliers, (c) drifting values, (d) co-linearity, and (e) sampling intervals. Missing values refer to the data values that have not been recorded and are not included in the datasets. Data outliers refer to the data values that lie an abnormal distance from other values and may be due to measurements errors, sampling problems, and natural variations. Drifting values are inaccurate measurement readings caused by factors such as contamination, vibration, or extreme temperature. Co-linearity refers to the correlation between input values. This phenomenon hinders statistics models from making accurate predictions. The sampling interval is the time between the measurements taken and data are recorded.

Kaedlec et al. also described a five-steps pipeline for the designing process of virtual sensors. The pipeline includes the steps of: (a) data inspection, (b) data selection, (c) pre-processing, (d) model selection—training—validation, and (e) sensor maintenance. The aim of data inspection is to gain an overview of the data structure and identify any obvious problems that may be handled at this initial stage. Data selection refers to the selection of the appropriate data features for the training and the inference of the ML model. In pre-processing, the data are transformed in such a way that it can be more effectively processed by the actual model. Model selection—training—validation includes the necessary steps to build and evaluate all the candidate models, in order to select the most accurate one. Due to the changes of the data and the dynamic environment, the sensors should be re-trained and tuned on a regular basis. This constitutes the maintenance of the sensors.

Furthermore, Kaedlec et al. discuss various ML models that constitute the core of virtual sensors. The proposed ML models are: (a) principal component analysis, (b) partial least squares, (b) ANNs, (c) neuro-fuzzy systems, and (d) support vector machines. Following the same pipeline, new researchers have continued the design and development of virtual sensors until today. Sun and Ge [30] present the applicability of deep learning for virtual sensors. They discuss various deep learning models, tricks, and frameworks, in order to help designers for the development of virtual sensors. In contrast, for the implementation of virtual sensors, in this work, ML regression methods are used. The models are divided into five main categories, and, to the best of our knowledge, it is the first time that such a comparative experimental study is provided.

### 2.6. Other Research Gaps of Virtual Sensors

Other research gaps that are not addressed in this paper regard the input and historical data, which are provided to the virtual sensors for the training stage and during their operation [31]. This includes the selection of the appropriate data sources, but also to examine the robustness of virtual sensors, in case they work in outliers and abnormal values [32]. Furthermore, the predictive models of virtual sensors should be capable of performing interpolation and trend simulation. This means that the prediction mechanism should be generalized, in case the input observations are significantly different from the historical data [25]. This is an important challenge, since the internal process may change, as well as the environment around the processes.

## 3. An IoT Platform for Virtual Sensors

This section has two purposes. First and foremost, the methodology and the required IoT platform to train and run the virtual sensors have to be described. Specifically, the goal is to provide a guideline to new researchers for developing a custom, lightweight, and extensible IoT platform for testing new ideas, based on data-driven models, such as our proposed virtual sensors. The reader is reminded that the virtual sensors are based on ML models that replace the physical sensors and require training historical data. This brings important design decisions that should be taken into consideration. Second, this section describes the various physical and software components included in our IoT platform, such as the RPis, DHT11 sensors, data persistence, data visualization, and the graphical user interface. Furthermore, it is important to discuss how these components are interconnected, providing the logical flow of our methodology. Last, but not least, the described IoT platform architecture works as a proof of concept for the use of virtual sensors. This means that the core of the architecture is kept simple, but extensible. Thus, the presented components can be replaced with counterpart components developed with different technologies, without affecting the other parts of the platform. As an example, if the database management system (DBMS) is replaced from MySQL to MongoDB, in the data persistence layer, the whole platform will continue to work smoothly.

### 3.1. Overview and Utility of the IoT Platform for Virtual Sensors

The proposed platform monitors the temperature and humidity physically or virtually in each room of a smart home. To achieve this functionality, the platform utilizes IoT devices, specifically RP is equipped with DHT11 sensors. The IoT platform can also include actuators, provide warnings and take intelligent decisions based on the monitoring data. In this paper, how the proposed IoT platform will use the monitoring data in different use cases is considered out of scope, but there are multiple applications in the literature that could benefit from our platform [33].

The proposed IoT platform is compared against a baseline approach. In the baseline approach, one physical IoT device is placed in each room and gathers all monitoring metrics in the IoT platform, following a centralized computing paradigm. This baseline approach incurs the following drawbacks and difficulties. First, smart homes and buildings often include many rooms, making the purchase cost for all the devices high. Next, the overhead of using RPis, their programming, the peripheral tools needed (such as SD cards, DHT11 sensor, etc.), their fault tolerance, and their maintainability are also increased proportionally to the number of the devices. In addition, the communication of multiple IoT devices uses bandwidth and generates interference that burdens the network, especially when there are frequent and continuous data transmissions. The aforementioned difficulties can be alleviated if, instead of RP being used to measure temperature and humidity in every room of the house, virtual measuring sensors built with ML could replace them. To sum up, the benefits of using the virtual sensors, instead of physical sensors, are to: (a) keep the number of devices low, (b) save energy, (c) decrease the bandwidth usage, and (d) minimize the cost of creating an IoT network with physical devices.

### 3.2. Logical Workflow

Figure 2 illustrates the operational logical flow of the proposed solution. As it can be seen, there are two pipelines. The first pipeline, aims to train the virtual sensors. This includes all the steps, beginning from the deployment of physical sensors of the IoT devices, until the construction of the ML models that will replace them. The second pipeline has as a goal to extract the inference data from the virtual sensors. In particular, this inference pipeline begins with the monitoring of one physical sensor, and it provides the physical measurements to the virtual sensors that are the output of the training pipeline, and eventually, the virtual sensors infer the temperature and the humidity in every room.

#### 3.2.1. Training Pipeline

In the training pipeline, two IoT devices need to be placed in two separate rooms. The first room is called the reference room, which has one physical IoT device that never moves, and it will not be replaced by the virtual sensors. In the second room, which is named the target room, a physical IoT device is placed that will eventually be replaced by the virtual sensor. Following, the data by these two IoT devices are monitored, creating two time series data. This process of monitoring two separate rooms is repeated for all the rest of the rooms, keeping the reference room constant, while the target room is variable.

Figure 3 depicts the smart home that will be used in the experimental evaluation. The bedroom is set as the reference room, where the permanent physical IoT device is placed. The temperature and humidity measurements of the reference room are depicted in Figure 3 using black fonts. The corresponding measurements of the virtual sensors in the target rooms are depicted with pink fonts. The monitoring data are depicted as time series plots in Figure 4 and Figure 5. Specifically, Figure 4 illustrates the time series metrics of the temperature and humidity in the reference room, while Figure 5 depicts the monitoring data of the second physical sensor in a target room that will be replaced by the virtual sensor.

The controller in the training pipeline monitors the pair of IoT devices and stores the measurements using time series structures in a database. It is important to mention that the two time series should be synchronized. They should begin at the same timestamp, and the sampling of the IoT devices will take place with a constant time step. The duration of the time steps is a configurable variable that depends on the use case. When a sufficient amount of data is gathered, a ML model is trained based on the historical data sequences. The ML model takes the reference room measurements as input and learns to predict the temperature and humidity of a target room. This process takes place sequentially for all the rooms, in order to train a virtual sensor for each of them.

#### 3.2.2. Inference Pipeline

In the inference pipeline, the controller takes the temperature and the humidity data from the reference room with a socket connection and provides them to the virtual sensors. Next, the predicted metrics go to a micro-service that visualizes them through a graphical user interface. The micro-service is extensible and can provide more functionalities, such as to manage other IoT devices, send signals to actuators, and make intelligent decisions based on different use cases. In the next subsections, more details for each component of the proposed IoT platform are provided.

### 3.3. Physical Sensor

The physical sensor that is used in the proposed IoT platform is the DHT11 module. The DHT11 module [34] features a temperature and humidity sensor complex, with a calibrated digital signal output. By using the exclusive digital signal acquisition technique and temperature and humidity sensing technology, it ensures high reliability and excellent long-term stability. This DHT11 module sensor consists of two parts, the DHT11 sensor and a module. DHT11 sensor includes a resistive-type humidity measurement component, a negative temperature coefficient (NTC) temperature measurement component, and a high performance 8-bit micro-controller, offering excellent quality, fast response, anti-interference ability, and cost-effectiveness. The module is actually a printed circuit board (PCB) with some required components, and, on top of that, it is placed the DHT11 sensor. The reason why this sensor is chosen is the convenience of use it offers, in conjunction with its low cost.

### 3.4. Raspberry Pi 3

RPi devices are characterized as low cost, fully customizable, programmable, portable size computer boards. These characteristics make them well-suited for IoT platforms. Their high connectivity capabilities also allow them to directly connect to other devices and applications through Wi-Fi, Bluetooth, and other access protocols. An additional physical interface with the outside world is the general purpose input–output (GPIO) connector. The GPIO is a 40-pin port that enables the RPi to communicate with physical devices such as sensors and actuators. In the proposed IoT platform, a DHT11 sensor is attached to the GPIO. Next, using the python libraries of RPi.GPIO (https://pypi.org/project/RPi.GPIO/ (accessed on 9 December 2022)) and dht11 (https://github.com/szazo/DHT11_Python (accessed on 9 December 2022)), and a python script was written that takes the temperature and humidity measurements.

RPis have the advantage, compared to other devices, such as Arduino, BeagleBone Black, Phidgets, and Udoo, and offer a very good trade off between computational capabilities and cost [35]. Specifically, Arduino, even if it is the cheapest device, has limited CPU power, cannot run more than one program at the same time, and lacks a full operating system. On the other side of comparison, Udoo has significantly better CPU power, but the cost can reach three times the cost of a RPi. In addition, RPi has a large community of users and developers that work on IoT projects, with better support and high contributions with a large number of public GitHub repositories. RPis can run various Linux and Windows distributions, but the Rasberry Pi operating system (previously called Raspbian) has been significantly tailored to the ARM micro-architecture and the particularities of the RPis. This gives the capability for one single device to receive data from the attached sensors, store and process them on device, and make decisions. Finally, it communicates data and decisions via our client-server IoT platform using a request–response messaging pattern.

### 3.5. Socket Programming

Socket programming offers a means of communication between applications using the transmission control protocol, following a server-client model. The server socket accepts connections to a specific port and IP address, while the client socket requests a connection. In this proposed IoT platform, the RPis with the DHT11 have the role of the server, while the machine that hosts the controller, the DBMS, and the virtual sensors has the role of the client. As it will be discussed in Section 3.9, the client can run on a Cloud VM, a node at the Edge, or another RPi.

The servers communicate via Wi-Fi to the client at a specific IP address and port. The client establishes a connection and receives the temperature–humidity measurements, along with their timestamps. The socket-based communication is a stateful protocol that facilitates the data storage. In the proposed framework, the client periodically sends a message to the server to get the temperature and humidity measurements—after, it decodes and splits them, and, finally, stores them in the database. It should be noted that other protocols could be used, such as MQTT, without changing the design of the proposed IoT platform. However, since in the proposed implementation, the Pi’s are relatively computationally strong devices and the power supply for the particular implementation is available, it was decided to use socket-based communication. For other types of applications, with more constrained IoT devices and less efficient access networks, more light-weight protocols, such as CoAP or MQTT, could be used instead.

### 3.6. Data Persistence

An IoT platform can handle a significant amount of data from devices and sensors, in order to take decisions and give real-time responses. The data should be stored on a disk, in case they do not fit in memory, for easy retrieval and further analysis. There are three main approaches for the data persistence based on the volume, the frequency of data, and the application requirements. The first option is to deploy a DBMS on the RPi [36]. This solution has the benefit that it improves the latency, as it eliminates the need to move data to cloud storage. It also improves the security and ensures the privacy protection. Multiple lightweight DBMS work well on RPis such as SQLite, MySQL, TinyDB, MariaDB, and PostgreSQL. In the proposed methodology, a MySQL DBMS was used, since it is appropriate for small projects that do not require much scalability and come with zero configuration. MySQL automates storing data in tables, following most of the SQL-92 standard, and it is characterized by the fact that it does not have a separate server process.

In case a larger database with more scalability, strong security, authentication features, and the ability of multiple users access is required locally, a DBMS running on a workstation at the Edge of the network will be a good solution. In case there is a need to store structured data, an SQL DBMS will be chosen, while for unstructured data, a non-SQL DBMS, such as MongoDB, is suitable. The time series temperature/humidity measurements can be stored either in tables in an SQL DBMS or in a time series collection in MongoDB. The third option is to connect the RPi with a cloud service, such as Microsoft Azure IoT, Google IoT Cloud, Amazon AWS IoT, Oracle IoT Cloud, and Cloud4RPi. In this case, the RPi sends sensor data to cloud services using a REST API, and the cloud services store and analyze the data, following a pay-as-you-go policy model. In addition, the cloud services can also control and send data to the RPis. This functionality has not been used on the platform, but can be a potential extension for a different smart home, smart city, or industrial use case.

For the needs of the proposed humidity/temperature use case, the measurements are stored and ordered by timestamps in different SQL tables. The SQL tables have the name of the room id in which the sensors are placed and where the measurements are taken. Moreover, the SQL queries to retrieve the data for the data visualization and the training of the virtual sensors have been written. Special emphasis has been given to retrieve the synchronized pairs of parallel measurements in two rooms for a variable time duration. Further data preprocessing services have been developed, in order to ensure the good quality of data, such as the cleansing of duplicate records, imputation of missing values, and an outlier detection mechanism.

### 3.7. Virtual Sensors

A virtual sensor is a software component that is based on a ML model and provides indirect temperature and humidity measurements. In the proposed use case, a smart home includes one physical sensor installed in a reference room and multiple virtual sensors corresponding to the rooms in which we want to monitor the temperature and humidity virtually. The virtual sensors work in two separate modes: (i) the training and (ii) the inference. In the training mode, the pairs of time series measurements are retrieved by the DBMS and train a ML model. A regression ML model learns the relationships between the reference room and the room, which is monitored virtually, with the objective of minimizing the errors in the forecast values. Every target room has a different relationship with the reference room that is determined by factors such as its geographic orientation, the type of wall insulation, and if it faces the sun and the wind. This relationship cannot be represented by a simple linear regression function, since the data observations may include non-linear and complicated patterns. After the training of virtual sensors is completed, the IoT platform moves to the inference mode. In the inference mode, the virtual sensors take the real measurements from the physical sensor in the reference room as input and estimate the temperature/humidity in the other rooms.

The regression models have two inputs that correspond to the temperature and humidity of the reference room. They also have one output that corresponds to the temperature or the humidity of the target room. However, a question arises as to what is the most appropriate regression ML model to use. In order to make this selection, the accuracy in the predictions, the inference time, the training time, and also the accuracy, as a function of the training data size, are under consideration [37]. This last point is an important factor that many times has been misunderstood by the researchers. It is a common belief in the literature that the performance is improved as the data size and the number of deep learning layers and neurons increase. This is true for very complicated tasks that traditional ML models lack sufficient capacity, compared with multi-layers perceptrons. However, this performance advantage comes with the disadvantage of high demands in computational resources in the training process. In the case that an IoT application runs at the Edge of the network, the computational resources are limited, let alone, in this proposed use case, where a RPi is used. Thus, if a lightweight ML model has satisfactory performance, it should be selected, instead of a deep learning model that is infeasible to be trained at the Edge. Section 5 presents various ML models and the methodology to build the virtual sensors around them.

### 3.8. Graphical User Interface

The proposed platform, in addition to making real and virtual measurements, must also be able to present the final results in an understandable way to its user. In essence, the virtual measurements that are carried out are displayed in detail to the final user of the platform, together with some other interesting historical data. The graphical user interface, in this case, is based on Flask Server. Flask is a small and lightweight Python web framework that provides useful tools and features that make it easy to build web applications in Python. Flask uses the Jinja templating engine to dynamically generate HTML pages using familiar Python concepts, such as variables, loops, lists, and so on.

Through the application, the user has access to the measurements made for each room (physical measurements for the reference room and virtual measurements for the target rooms) in real time. Additionally, the user has the opportunity to see the historical data, along with detailed information regarding the average, max, and min values of the temperature and humidity for each day, since the start of the measurement period, as depicted in Figure 4 and Figure 5. Specifically, a graph presents the change of the average temperature and humidity for the rooms in question over time, using the Matplotlib library, which is a comprehensive library for creating static, animated, and interactive visualizations in Python. These data can be of great importance, if they can be used with other applications of a smart home, such as smart metering, HVAC applications, etc., to create a better living environment, while making energy and cost efficient decisions.

### 3.9. Architecture of the Platform

The hardware and software components, described in the previous subsections, cooperate with each other and constitute the final IoT platform illustrated in Figure 6. The physical temperature and humidity readings from the RPi are sent to a processing Edge node through socket programming. Next, they are being prepossessed and stored in the local database. The processing Edge node can also be a RPi or a workstation computer. In the proposed implementation, a RPi is used to prove that the platform is lightweight and can be executed and deployed flexibly in cheaper and smaller devices. The IoT platform also includes the ML algorithm for the virtual sensors, which will be used for model training based on the stored data. Which ML algorithm is appropriate for a temperature/humidity virtual sensor is a research question that is discussed theoretically in Section 4 and answered experimentally in Section 5.

As explained in Section 3.2, the IoT Platform can work in two different states, the training and the inference. The controller determines which of them will be activated based on the interaction of the user with the GUI, the connected RPis, and the available data in the database. In order to run the training, it requires an efficient amount of data pairs in the database. In a similar way, in order to run the inference, it requires the train to be completed. The controller makes these checks and runs the appropriate pipeline components as explained in Section 3.2. Last, but not least, it is important to clarify that, in this work, the various types of protocols that can be used and the various security threats that can arise are not examined. In case the reader is interested in these topics, the following papers are suggested [3,38].

## 4. Utilizing Machine Learning for Virtual Sensors

The virtual sensors are implemented based on ML models that capture the relationships between the reference room and a target room. In the training stage, historical data are provided to the ML models, in order to learn the temperature/humidity values of the target rooms based on the temperature/humidity values of the reference room. The training process builds a model that minimizes the error between the predicted and true values of the target variable through an iterative optimization process. After the training of the virtual sensor is completed, only the sensor values of the reference room can be used to infer the temperature and humidity values of the target rooms.

Using two DHT11 sensors, one in the reference room and one in the target room, the temperature/humidity values are recorded and the training dataset is constructed. For every target room, two separate regression models are trained, one for the humidity and one for the temperature. The example smart home, depicted in Figure 3, has six distinct living areas. A DHT11 sensor is placed in the bedroom, which stands as the reference room, and records the temperature/humidity with a constant interval. Next, there are five virtual sensors, with each one corresponding to a target area for the prediction of the temperature and five virtual sensors for the humidity. Predicting the future value of temperature or humidity is a regression task because the value is a continuously valued attribute. This is an important characteristic of the data that is taken into account for the selection, comparison, and evaluation of the potential ML models that will be presented in this section.

### 4.1. Time Series Sensor Metrics and Dataset Construction

The historical data used for the training of the ML models constitute of a set of time series. A time series is defined as a series of values of a quantity obtained at successive times, with equal intervals between them. In the context of a smart home, the temperature and humidity conditions of each room are sampled with a constant time interval. It is important to formulate parallel and synchronized measurements in every pair of rooms, where one room (i.e., bedroom) always stands as a reference room and does not change. The sensor measurements constitute data points indexed in time order, and for every pair of rooms, there are four time series, which are the following:Reference room temperature;Reference room humidity;Target room temperature;Target room humidity.

While the historical data have a time series structure, the inference process of the virtual sensors does not follow a statistical time series forecasting methodology, but a regression ML. This is something that has to be explained. Statistical time series models, such as ARMA and its variations, such as ARIMA and SARIMA, are based on the stationary assumption, which means that its statistical properties do not change over time and use an auto-regressive process to forecast the next value as a linear combination of past values of the target variable. Thus, while the sensor metrics are recorded and presented as time series, the classical time series forecasting models cannot capture the relationship between different variables and, specifically, the relationship between the reference and the target metrics. Regression ML models estimate the relationship between a dependent variable and one or more independent variables, as will be discussed in the following subsection.

### 4.2. Regression Models

Regression analysis is the process of predicting the value of one continuous target variable as a function of one or more input variables. In the proposed use case, this takes place by training a model to map the independent variables, which are, jointly, the temperature and humidity of the reference room, to the dependent variable of temperature for the one virtual sensor and the humidity for the other virtual sensor. The type of relation between the dependent and the independent variables is not known beforehand, and it is a topic of research. The relation of the temperature of the target room to the temperatures and the humidity of the reference room can depend on various factors, such as their geographic orientation, the type of wall insulation, and whether they face the sun or the wind. How these factors affect the relation of the temperature and humidity between rooms is a very difficult problem to express with a numerical equation. This is the reason why non-linear, or even black-box data-driven, solutions are chosen. The main categories of regression ML models are presented in the following subsections. At this point, it is necessary to provide a brief overview of the regression ML models. In the next section, the models will be experimentally evaluated, in order to conclude to the most accurate for the virtual sensors use case. If the reader wants an extensive study of regression ML models, a reference to the Christopher M. Bishop’s book *Pattern Recognition and ML* is suggested [39].

#### 4.2.1. Linear Models

The simplest and most popular regression approach is linear regression. The general form of linear regression defines the target or dependant variable as a linear function of the independent or input variables. The advantage of linear regression, compared to other methods, is that it has good analytical properties, but it does not perform well in many natural phenomena that cannot be expressed by linear functions. Equation (Equation 1) gives the temperature of the target room $y_{temp}^{target\_room}$ as a linear function of the temperature in the reference room $y_{temp}^{reference\_room}$ and the humidity of the reference room $y_{hum}^{reference\_room}$. The coefficients/weights $w_{temp0}$, $w_{temp1}$ and $w_{temp2}$ are learnable parameters that fit on historical data to minimize the residual sum of squares between the true values of the observations and the predicted ones. In the same way, Equation (Equation 2) gives the humidity in the target room $y_{hum}^{target\_room}$ as a linear function of the $y_{temp}^{reference\_room}$ and the $y_{hum}^{reference\_room}$ using different coefficients $w_{hum0}$, $w_{hum1}$, and $w_{hum2}$. These coefficients express the relation of the input variables with the target room humidity, instead of the temperature.
(1)$$y_{temp}^{target\_room}=w_{temp0}+w_{temp1}\cdot y_{temp}^{reference\_room}+w_{temp2}\cdot y_{hum}^{reference\_room}$$
(2)$$y_{hum}^{target\_room}=w_{hum0}+w_{hum1}\cdot y_{temp}^{reference\_room}+w_{hum2}\cdot y_{hum}^{reference\_room}$$

In case that the temperature and humidity of the reference room are highly correlated, the phenomenon is named multicollinearity. Multicollinearity renders the ordinary least square method for the estimation of temperature $wt_{0}$, $wt_{1}$, and $wt_{2}$ and humidity $wh_{0}$, $wh_{1}$, and $wh_{2}$ coefficients inefficient. Ridge regression [40] addresses the multicollinearity by imposing a penalty on the magnitude of the coefficients. The minimization of the penalized residual sum of squares converges to more optimal coefficient configurations for the regression model. In the case that the priors for the coefficients are given by a spherical Gaussian, there is the Bayesian ridge regression model [41]. Ridge regression can be effective, even if there are non-linear data observations, by using the kernel trick [42]. Kernel ridge regression performs ridge regression with a potentially infinite number of nonlinear transformations of the independent variables. ElasticNet [43] is also an extension of ridge regression that includes variable selection and regularisation, in order to improve the accuracy. Last, but not least, instead of using the ordinary least square method for the training of the regression coefficients, alternative optimization methods, such as stochastic gradient descent (SGD), can be used [44].

#### 4.2.2. Support Vector Regression

Support vector regression (SVR) [45] is an effective regression model for real-value problems, especially when the historical data are sparse. Sparse data means that the number of data points is small, compared to the number of input variables. SVR has also good performance, with non-linearly separable data observations using slack variables. In addition to this, if the historical data observations are not characterized by linear decision boundaries, they can be mapped to a higher dimensional space and apply SVR using the kernel trick. The training of SVR takes place with a symmetrical loss function, which equally penalizes high and low prediction errors.

SVR fits hyperplanes in an n-dimensional space and provides predictions, with function *f*, as given in Equation (Equation 3), based on a decision boundary and an error margin ϵ. The *f* represents the temperature or humidity in the target room. The learning process tunes the ϵ to gain an acceptable accuracy, minimizing the coefficients *w* and satisfying the constraints in Equation (Equation 4). *w* is a feature vector including the learnable parameters of the hyperplanes, and *x* is a feature vector that includes the temperature and humidity of the reference room. The learnable parameters of SVR are different from the learnable parameters of the linear models because they follow different ML methodologies and they are based on different mathematical models. Equation (Equation 4) can be solved using the Lagrange multiplier method.
(3)f(x)=〈w,x〉+bwithw∈X,b∈R
(4)$$\begin{array}{c} minimize\;\;\;\frac{1}{2}{∥w∥}^{2}\\ subject\;to=\left\{\begin{array}{c} y_{i}-\langle w,x_{i}\rangle -b\leq \epsilon \langle w,x_{i}\rangle +b-y_{i}\leq \epsilon \end{array}\right. \end{array}$$

SVR is promising, in terms of generalizing unforeseen data observations. This has practical applications for virtual sensors working in unusual weather phenomena that differ from the conditions the historical dataset constructed. On the other hand, SVR has some significant limitations that make it unsuitable as the basis for virtual sensors. The limitations are that: (a) SVR using the kernel trick can be quite sensitive to overfitting, (b) it is affected by noisy data, and (c) it fails to adapt in data observations with close input feature vectors, but significantly different outcomes.

#### 4.2.3. Bootstrap Aggregating

A ML regression approach that often performs very well is ensemble learning. Ensemble learning is based on the principle of combining multiple base learners to form one strong learner. The two ensemble learning categories that are examined for virtual sensors are the bootstrap aggregating and boosting methods. Bootstrap aggregating [12], also named the bagging process, often uses decision trees as base learners. The decision trees are trained on separate variations of the original dataset, which are called bootstrapped data sets. The bootstrapped data sets are generated by sampling with the replacement of the original dataset of temperature and humidity. Every individual base learner in the ensemble has a different perspective of the prediction task, promoting a diversity that removes the variance of the single base learners and is less prone to overfitting. In the end, the ensemble’s overall assessment is considered by the aggregation of the individual outputs.

One well-established implementation of bootstrap aggregating method is the random forests [46]. Random forests are based on multiple decision trees trained on bootstrapped data sets. Decision trees are base learners that rely on a tree-like data structure and hierarchically partition the feature space using a sequence of tests on the individual features. Decision trees have limitations, for instance, they memorise irregular data patters and overfit the historical data. Random forests aggregating multiple decision trees overcome these limitations and improve the robustness and predictive performance. The average predictions of the individual decision trees is the final output of the random forests.

#### 4.2.4. Boosting

Boosting [13] has a common characteristic with bootstrap aggregating that works in different variations of the historical dataset. The main difference is that, in bootstrap aggregating, the variations of the datasets are generated by sampling with a uniform distribution, while in the boosting methods, the miss-predicted data instances have higher probability to be sampled, compared with the correctly predicted instances. Therefore, there are sequential rounds of dataset constructions and individual base learners training. In the first variation of the dataset, all data instances have equal sampling probability. From this dataset, the first base learner is trained and evaluated. Next, the second variation of the dataset is generated with a sampling probability proportional to their evaluation errors. The process of data generation, the training of the base learners, and the evaluation continue until the evaluation error has been shrunk under a given threshold value or a specific number of base learners is added to the ensemble. In the inference stage, the output of the boosting method is the weighted average of all individual base learners.

Adaptive boosting (AdaBoost) [47] is one of the most widely used boosting algorithms, with applications in numerous practical use cases. AdaBoost iteratively uses decision stumps in separate features, following the principle that each subsequent decision stump is forced to concentrate on data instances that are significantly miss-predicted by the previous ones in the sequence. Decision stumps are one-level decision trees with only one split. Gradient boosting [48] goes a step further, using an optimization process on a differentiable cost function to choose which decision stump will be added in the ensemble. Specifically, gradient boosting applies the gradient descent algorithm on (a) squared error, (b) absolute error, or (c) huber loss between the predicted values and the target values. Since the training of the virtual sensor takes place on an IoT device, it is crucial to resort in lightweight boosting methods, such as LightGBM (LGBM) [49] and Histogram-based gradient boosting [50]. These methods significantly speed up the training process by reducing the number of values for continuous input features. This takes place by discretizing or binning the temperature and humidity instances into a fixed number of buckets. Last, but not least, it is mentioned that XGBoost (XGB) [51] has won many prestigious ML competitions and has many improvements, compared to other boosting methods, as it penalises the complexity on trees, shrinks the leaf nodes, uses feature selection, and applies the Newton method, instead of a gradient descent.

#### 4.2.5. Regression Artificial Neural Network

Regression artificial neural networks (RANN) [14] are based on hierarchical representations and transformations of the input feature vector, in order to predict a continuous output variable. The representation of the input vector takes place by a number of neurons that apply a non-linear activation function, such as ReLu and softmax to the input. The artificial neurons are organized in sequential layers, forming a directed graph from the input feature vector to the output continuous value. In case the hidden layers between the input and the output are many, these models are named deep learning. Regression deep learning models include many design decisions and became known as hyperparameters, and they should be taken into consideration, in order to provide accurate results. Some of the hyperparameters are the number of layers, the number of neurons in each layer, the activation function, the optimization process, the loss function, and the percentage of regularization. A hypertuning process, such as Bayesian optimization or a genetic algorithm, can automatically search through many hyperparameter combinations, in order to conclude to a close-to-optimal RANN.

The literature mentions different types of RANN, categorized according to the type of problem they tackle and the data representation they use. Specifically, recurrent neural networks are used for sequential data values and convolutional neural networks for grid-like topologies. The prediction of temperature and humidity in a target room works as a function of the sensor measurements in the reference room. In this process, a feature vector representation and a feed-forward neural network are used. Lately, ANNs have gained massive popularity and, indisputably, the majority of researchers conclude that they outperform other ML and regression models [52]. The large number of parameters coded as weights and biases in the synapses of neurons give plasticity to the models for efficient adaptation and generalization to complicated tasks and data. This takes place without memorization and without overfitting the historical observations.
(5)$$F\left(x\right)=w_{3}^{T}\cdot \sigma \left(w_{2}^{T}\cdot \sigma \left(w_{1}^{T}x+b_{1}\right)+b_{2}\right)+b_{3}$$

The hypertuning process concluded a feed-forward RANN with three layers, as given in Equation (Equation 5), where σ is the ReLu activation function given as σ(x)=max(0,x), T denotes the transpose matrices of the neurons weights $w_{1},w_{2},w_{3}$ of each layer, and $b_{1},b_{2},b_{3}$ are the neurons biases of the layers

## 5. Implementation of the IoT Platform and Experimental Evaluation

The IoT Platform has been implemented, as described in Section 3. During the operation of the platform, the historical data are recorded, following the training pipeline, as depicted in Figure 2. Following, the ML models that constitute the core of the virtual sensors are built using the recorded historical data. In Section 5.1, the front-end of the IoT platform is presented, as displayed by the end user. In Section 5.2, the experimental evaluation and comparison of the various ML models are provided. These models are described in Section 4. The ML literature includes so many regression models that it is not feasible to examine them all. Fourteen well-established regression models are implemented and compered, in order to conclude which is the most accurate for the virtual sensors of temperature and humidity. In addition, the pros and cons of each regression algorithm, based on the target metric, the room, the magnitude, and the frequency of errors, are discussed. This experimental comparison to select the regression ML model that will be used for the virtual sensors took place only one time in the lab. Thus, it is possible to observe the applicability of the proposed methodology and select the most accurate model.

### 5.1. Implementation of the IoT Platform

In Figure 3, we have seen the web GUI depicting the smart home where the experiments took place. The bedroom was set as the reference room, and the last physical measurements of temperature and humidity are indicated with black fonts. Additionally, these physical sensor values are provided to the virtual sensors, and the inferred temperature/humidity values for the remaining rooms are depicted with pink fonts. For more information about the historical data, the user, by clicking on the *Historical Data* button on the page, is transferred to the HTML page of ‘Historical Data’. On this page, and as can be seen in Figure 7, there is information about the latest update from the physical sensors. There are also graphical representations of the data for the reference and the target measurements depicted in Figure 5. These historical metrics are used in the training stage, in order to build the virtual sensors. In addition, the platform can provide basic statistical measurements of the physical or virtual sensors on a daily basis, as shown in Figure 8.

It is important to mention that the information and the GUI layout presented may not be in the interest of end users looking for a commercial application. The proposed IoT platform [53] is designed and developed as a roadmap for researchers that need a lightweight, simple, extensible, and cost-effective alternative to commercial IoT platforms. This platform could help the researchers to integrate and test their innovative ideas in the domain of a smart home or a local IoT environment emphasizing the use of physical and virtual sensors. The quality of experience was good, with a response time of less than 180 ms for all the services it offers, while the communication between the DHT11 equipped RPis was established successfully via Wi-Fi through the socket connection.

### 5.2. Experimental Comparison of Regression Models

The virtual sensors have been implemented and experimentally evaluated in the Python 3 programming language using the libraries NumPy, pandas, Scikit-learn, SciPy, and Keras. The environment that was used for the experimental comparison was a Jupyter notebook of the Google Colaboratory, while the final Python scripts were run on the RPi devices.

#### 5.2.1. Evaluation Metrics

In the provided experiments, Celsius degrees were used as the unit of measurement for temperature, as well as the relative humidity for humidity. For the evaluation and comparison of the regression ML models, the mean absolute error (MAE) and the mean squared error (MSE) evaluation metrics were used.

The MAE represents the difference between the original (real) and predicted (pred) values extracted from the mean of the absolute difference in the data set.
$$MAE=\frac{1}{n}\sum_{i=1}^{n}\left|y_{i}^{real}-y_{i}^{pred}\right|$$

MSE represents the difference between the original and predicted values extracted by the square of the mean difference in the data set.
$$MSE=\frac{1}{n}\sum_{i=1}^{n}{\left(y_{i}^{real}-y_{i}^{pred}\right)}^{2}$$

As MSE assigns a higher weight to larger prediction errors, it is more useful when large prediction errors are undesirable. However, MAE is preferred when all errors have the same importance. Furthermore, MSE is equal to or greater than MAE and may increase more than MAE as the dataset increases.

#### 5.2.2. Experimental Outcomes

For the experiments, multiple time series were recorded, which include synchronized pairs of temperature and humidity of the reference room and one of the target rooms, as depicted in Figure 3. Every time series has a one-minute time step and is sampled over one week. The time series is split into a training set containing the first 66% part of the sequential observations and the testing set including the last 34% of the observations. For every virtual sensor, a separate ML model was used, trained from scratch, without applying any transfer learning method among the models.

In Table 1, the experimental evaluation results were summarized, in terms of MAE and MSE for the linear regression, the SVR, bootstrap aggregating, boosting, and RANN models that are described in Section 4. Bold fonts is mark the best model in every target room. In the case that two models have the same performance, they are both marked with bold fonts. Data scientists and ML researchers, when they make an experimental comparison of different models, prefer to present one model that beats the others and has the best performance. While this simplistic approach has many benefits, since it makes for straight-forward data storytelling extolling and proposing one model that comes as an answer to the research challenge, it is far from reality for many practical and daily challenges.

By delving into the results of Table 1, it can be observed that the virtual sensors perform very well, having a MAE close to one in most of the rooms and ML models. These MAE values are indicative that, most of the time, the virtual sensor outputs are very close to the real values. This conclusion will become more evident when the outcomes are discussed in the next tables. It is to be noted that the MSE can not express what the potential worst errors of the models are. However, it states which models should be avoided, in the case that large errors are undesirable, even if they occur rarely. As it can be seen, in most cases, the models that output small MAE also output small MSE. There is not one model that has the best performance in all rooms, but random forest excels because it has the smallest MAE in three of the five target rooms and the second best in the remaining two.

Table 2, Table 3, Table 4, Table 5 and Table 6 show the percentage of times that the ML models predicted exactly the correct temperature and humidity and the percentage of times the predictions had an error less than or equal to one degree for every target room. Again, it is understood that, even if there is high accuracy, there is not one specific ML model that outperforms the others. Nor yet, is there one category of regression models that performs better than the others, since bootstrap aggregating, boosting, and linear models equally have the first position in some circumstances. Nevertheless, it is obvious that the exactly correct prediction of temperature varies in a range from 58.8% to 90.91%, and for all the rooms, the prediction with an error equal or less to 1% ranges from 84.15% to 100%. These numbers give us confidence that, almost always, the correct temperature will be predicted, and if the virtual sensor fails to give a correct measurement, it will deviate for less than 1%, which is a smaller error than the accuracy of the DHT11 sensor. The accuracy of DHT11 sensor is ±2 °C in the temperature range of 0–50 °C and ±5% for the humidity in the range 20–90%.

Another observation is that the accuracy for the humidity is significant lower than the temperature. The reason behind this behavior is that the range of values for the humidity is greater than that of temperature. Nonetheless, such deviation is insignificant in practical applications and not easily perceived by humans. The exactly correct prediction of humidity is in a range from 12.27% to 51.82%, and for all the rooms, the prediction with an error is equal or less to 1% ranges from 52.65% to 87.58%. One more remark is that, many times, there are two, three, or more regression models that have the best accuracy. This can be attributed to the fact that the data observations include repeated patterns, and the models that capture these data patterns have significantly better performance than the models that do not.

The virtual sensors should respond with low latency, in order to provide timely results with no significant computational and transmission delays. Regarding the transmission time of the data, it is as fast, or even faster, than physical sensors, since they run on the same device that hosts the IoT Platform, eliminating any propagation delays or delays from retransmissions due to weak channel conditions. Regarding the computational delay, it can be distinguished in two different types, the inference time and the training time. The inference time is defined as the time overhead, since the data came into the regression model until the result was extracted. The inference time for one single prediction in the ML models ranges from 0.8 ms to 40 ms, and for the RANN, it is close to 48 ms. The training time is defined as the time it takes a model to fit its parameters, also known as learning from historical data. The training time in ML models for one day historical data ranges from 2 ms to 817 ms, and for the RANN, it is close to 172 s. A distinction is made between RANN and the other ML models because RANN includes the hypertuning algorithm, and, generally speaking, deep learning methods are notorious for being computationally expensive. In any case, the training occurs sporadically when the virtual sensors need updating, and these training times should not be of concern, even if the processing takes place in a RPi.

The last comment is about the importance of selecting the correct ML model. The results in Table 3, Table 4, Table 5 and Table 6 show that the selection of the ML model is a very important factor for the good performance of the virtual sensors. As an example, in the target area of the road balcony, the selection of ElasticNet gives an accuracy of 13.03%, while the selection of AdaBoost gives an accuracy close to 99.50%. For this reason, it can be deduced that the experimental comparison of the different ML models, even though they provide different results for every target area, should be carried out, since it gives the most important answer to the question of how to design a virtual sensor. This does not necessarily imply that the researchers have to make tedious experiments and comparisons every time they need a new virtual sensor. In contrast, an AutoML [54] approach can be applied that eliminates any human intervention from the training and evaluation process and automatically concludes which is the most accurate ML model.

## 6. Conclusions and Future Work

This paper discusses how we can design and implement virtual sensors in an IoT environment, in the context of a smart home use case. First, a guideline of how researchers can build their own custom IoT platform was provided, in order to test and develop a prototype virtual sensor. Next, a detailed analysis was laid out of how ML models can learn the relationships between the metrics of physical sensors, in order to replace most of them with virtual counterparts. The proposed methodology has been implemented and evaluated in a real environment, while the experimental outcomes confirmed the applicability of our approach.

As future work, it is worth investigating the use of open data to improve the performance of the proposed model or completely replace the physical sensors. It is important to examine whether models can be built that take input of open weather APIs and learn to predict the temperature and humidity in every room. Currently, virtual sensors are used only for monitoring purposes. The aim is to manage heating/cooling devices and integrate physical actuators together with virtual sensors. The ultimate vision of this future direction is to bring us closer to an ambient intelligent environment that automatically makes decisions and interacts with humans.

## Figures and Tables

**Figure 1 sensors-23-01328-f001:**
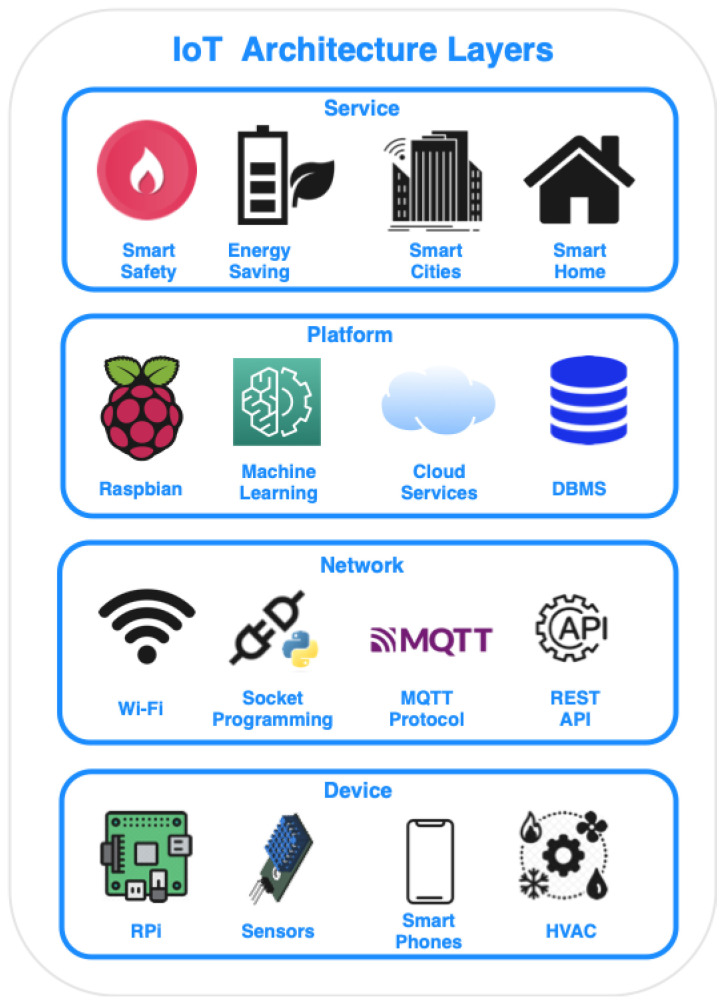
The four layers of IoT structure.

**Figure 2 sensors-23-01328-f002:**
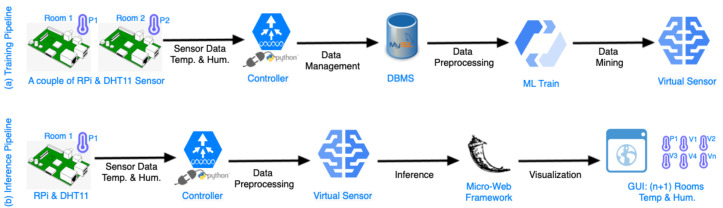
Logical flow of training and inference IoT virtual sensors.

**Figure 3 sensors-23-01328-f003:**
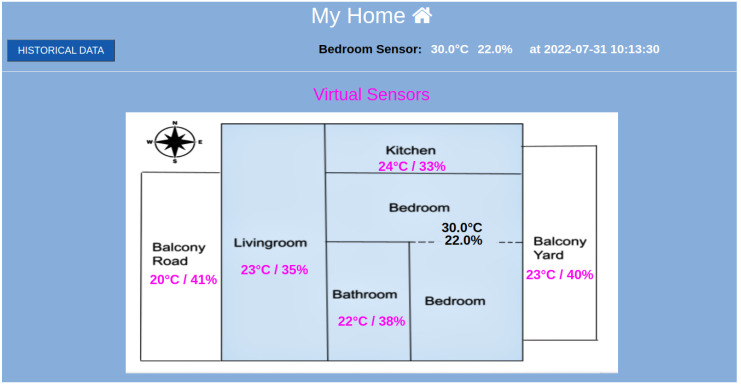
Physical and virtual temperature and humidity in a smart home.

**Figure 4 sensors-23-01328-f004:**
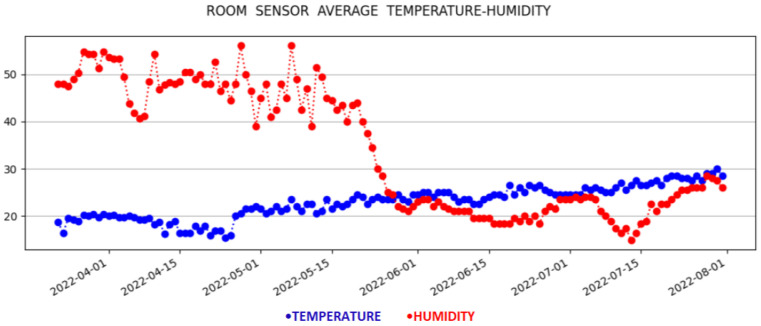
Temperature and humidity time plots in reference room.

**Figure 5 sensors-23-01328-f005:**
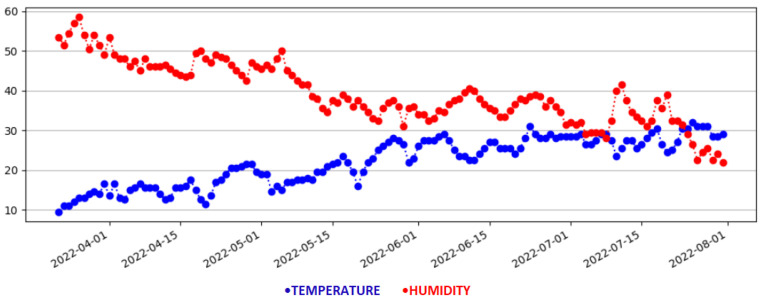
Temperature and humiditytime plots in a target room.

**Figure 6 sensors-23-01328-f006:**
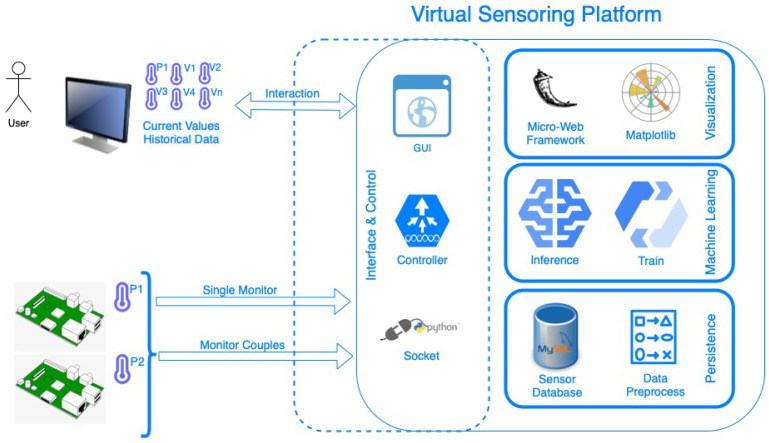
Architecture of the IoT platform.

**Figure 7 sensors-23-01328-f007:**
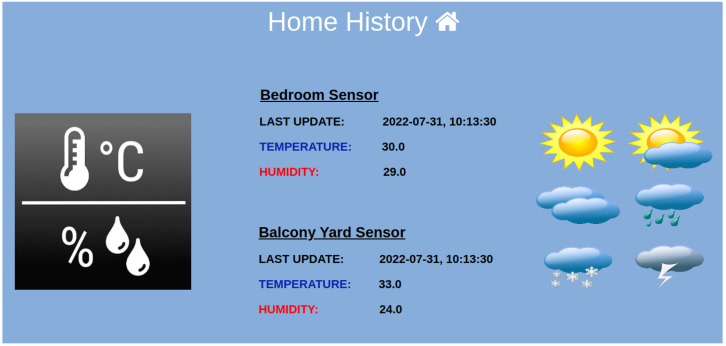
Latest temperature/humidity update from indoor and outdoor sensor.

**Figure 8 sensors-23-01328-f008:**
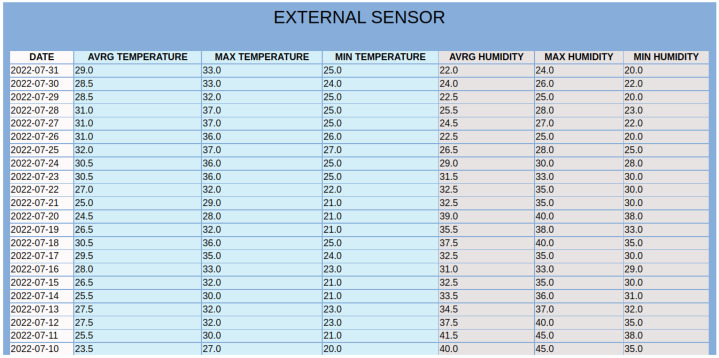
Statistical measures of temperature and humidity on a daily basis.

**Table 1 sensors-23-01328-t001:** Evaluation of algorithms for every room.

	Balcony Road		Balcony Yard		Bathroom		Kitchen		Living Room	
Method	MAE	MSE	MAE	MSE	MAE	MSE	MAE	MSE	MAE	MSE
AdaBoost	1.5979	7.9375	2.1441	11.4992	1.2489	4.6465	1.4622	5.6672	0.4514	0.5641
BayesianRidge	2.3006	13.8466	3.5628	31.6905	2.3008	15.9389	2.615	18.2141	0.6203	0.9227
CatBoost	1.7342	8.4643	2.7629	19.6283	1.5183	7.1468	1.6053	6.222	0.7497	1.5373
ElasticNet	3.2091	26.8315	6.1497	53.2716	3.7351	29.7033	3.1341	20.6357	1.5846	6.148
GradientBoosting	1.2766	**6.7852**	1.4735	**7.8794**	0.9742	3.286	1.175	4.4579	**0.403**	0.5164
HistGradientBoosting	1.2739	6.7954	1.4445	8.038	0.9616	3.3028	**1.1699**	**4.4341**	0.4086	0.527
KernelRidge	2.7284	17.9775	4.6942	36.4387	3.5894	29.4789	3.0437	20.3099	1.1785	2.1707
LinearRegression	2.3017	13.8427	3.5601	31.7293	2.2997	15.9374	2.6053	18.1556	0.6202	0.9223
LGBM	1.2739	6.7954	1.4445	8.038	0.9612	3.2955	**1.1699**	**4.4341**	0.4086	0.527
RandomForest	**1.2687**	6.81	1.4354	8.0045	**0.9494**	**3.2712**	1.1796	4.4817	**0.403**	**0.5159**
SGD	3.0123	23.4168	5.7962	46.7601	3.6078	26.5266	3.0486	19.0202	1.4105	4.266
SVR	4.7412	41.0185	6.3415	57.2641	3.8512	37.7262	3.2094	20.6586	1.6955	6.2129
XGB	1.6581	8.7605	2.4448	13.9481	1.0882	3.8424	1.1766	4.4844	0.4239	0.5275
RANN	1.4645	7.5782	**1.4235**	9.0569	1.3114	6.0497	1.2715	5.2777	0.4498	0.5973

**Table 2 sensors-23-01328-t002:** Evaluation of algorithms for the measurements bedroom-road balcony.

Balcony Road	% of Correct Value		% of ±1 Value	
Method	Temp.	Hum.	Temp.	Hum.
AdaBoost	**59.81%**	22.51%	**99.50%**	41.48%
BayesianRidge	21.83%	0.7%	68.66%	21.83%
CatBoost	26.41%	10.92%	66.2%	44.01%
ElasticNet	1.06%	0.0%	13.03%	0.35%
GradientBoosting	50.7%	24.3%	83.1%	62.68%
HistGradientBoosting	53.52%	**30.99%**	83.1%	69.01%
KernelRidge	1.06%	0.35%	4.93%	17.25%
LinearRegression	21.83%	0.7%	68.66%	21.83%
LGBM	53.52%	**30.99%**	83.1%	69.01%
RandomForest	54.23%	27.82%	82.75%	67.25%
SGD	1.41%	0.0%	5.28%	0.7%
SVR	13.38%	0.0%	40.49%	0.35%
XGB	30.99%	14.79%	71.83%	39.08%
RANN	58.80%	29.58%	84.15%	**69.37%**

**Table 3 sensors-23-01328-t003:** Evaluation of algorithms for the measurements bedroom-yard balcony.

Balcony Yard	% of Correct Value		% of ±1 Value	
Method	Temp.	Hum.	Temp.	Hum.
AdaBoost	32.75%	15.49%	73.24%	45.42%
BayesianRidge	21.83%	0.7%	68.66%	21.83%
CatBoost	26.41%	10.92%	66.2%	44.01%
ElasticNet	1.06%	0.0%	13.03%	0.35%
GradientBoosting	50.7%	24.3%	83.1%	62.68%
HistGradientBoosting	53.52%	**30.99%**	83.1%	69.01%
KernelRidge	1.06%	0.35%	4.93%	17.25%
LinearRegression	21.83%	0.7%	68.66%	21.83%
LGBM	53.52%	**30.99%**	83.1%	69.01%
RandomForest	54.23%	27.82%	82.75%	67.25%
SGD	1.41%	0.0%	5.28%	0.7%
SVR	13.38%	0.0%	40.49%	0.35%
XGB	30.99%	14.79%	71.83%	39.08%
RANN	**58.80%**	29.58%	**84.15%**	**69.37%**

**Table 4 sensors-23-01328-t004:** Evaluation of algorithms for the measurements bedroom-bathroom.

Bathroom	% of Correct Value		% of ±1 Value	
Method	Temp.	Hum.	Temp.	Hum.
AdaBoost	**89.57%**	6.75%	98.47%	45.09%
BayesianRidge	87.42%	3.07%	99.08%	25.77%
CatBoost	87.42%	5.52%	98.77%	39.26%
ElasticNet	63.19%	1.53%	98.16%	6.13%
GradientBoosting	82.82%	7.36%	99.69%	59.82%
HistGradientBoosting	82.52%	**12.27%**	99.69%	**60.12%**
KernelRidge	27.91%	0.61%	57.36%	10.43%
LinearRegression	87.42%	3.07%	99.08%	25.77%
LGBM	82.52%	**12.27%**	99.69%	**60.12%**
RandomForest	82.52%	**12.27%**	99.39%	**60.12%**
SGD	34.05%	3.99%	99.39%	9.51%
SVR	34.97%	4.29%	**100.0%**	15.34%
XGB	83.44%	10.12%	99.69%	51.23%
RANN	87.42%	**12.27%**	98.16%	52.76%

**Table 5 sensors-23-01328-t005:** Evaluation of algorithms for the measurements bedroom-kitchen.

Kitchen	% of Correct Value		% of ±1 Value	
Method	Temp.	Hum.	Temp.	Hum.
AdaBoost	72.44%	10.95%	99.29%	39.22%
BayesianRidge	68.2%	14.13%	**100.0%**	26.86%
CatBoost	**73.14%**	7.77%	99.29%	32.86%
ElasticNet	70.67%	0.35%	99.29%	1.06%
GradientBoosting	**73.14%**	15.9%	99.29%	51.94%
HistGradientBoosting	72.44%	15.19%	99.65%	**52.65%**
KernelRidge	22.61%	12.01%	60.78%	32.86%
LinearRegression	68.2%	13.78%	**100.0%**	25.09%
LGBM	72.44%	15.19%	99.65%	**52.65%**
RandomForest	72.44%	15.9%	99.65%	51.94%
SGD	39.93%	0.35%	86.22%	7.77%
SVR	22.61%	0.35%	93.99%	1.06%
XGB	**73.14%**	17.67%	99.29%	51.59%
RANN	72.44%	**26.15%**	99.29%	50.18%

**Table 6 sensors-23-01328-t006:** Evaluation of algorithms for the measurements bedroom-living room.

Livingroom	% of Correct Value		% of ±1 Value	
Method	Temp.	Hum.	Temp.	Hum.
AdaBoost	**90.91%**	46.67%	**100.0%**	86.06%
BayesianRidge	81.82%	33.64%	**100.0%**	76.67%
CatBoost	88.18%	28.18%	**100.0%**	66.97%
ElasticNet	80.91%	8.18%	**100.0%**	26.06%
GradientBoosting	88.18%	**51.82%**	**100.0%**	**87.58%**
HistGradientBoosting	88.18%	50.91%	**100.0%**	87.27%
KernelRidge	17.27%	24.85%	74.85%	70.91%
LinearRegression	**90.91%**	33.64%	**100.0%**	76.67%
LGBM	88.18%	50.91%	**100.0%**	87.27%
RandomForest	88.18%	**51.82%**	**100.0%**	**87.58%**
SGD	68.48%	8.18%	98.18%	33.94%
SVR	80.91%	8.18%	**100.0%**	26.06%
XGB	88.18%	51.21%	**100.0%**	**87.58%**
RANN	90.61%	49.09%	**100.0%**	83.33%

## Data Availability

Not applicable

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
