# Peer review of "Enabling Artificial Intelligent Virtual Sensors in an IoT Environment"

_sensors, 2023, doi:10.3390/s23031328_

Round 1
Reviewer 1 Report
In this article, the authors experimentally evaluate the performance of virtual sensors. The authors developed an IoT platform that runs on a
Raspberry Pi and equipped with physical temperature and humidity sensors. The idea of the article is very interesting. The authors provide a detailed description of the IoT platform. However, the article is a formal document; the authors must review the whole document. The authors should avoid using " ' " and " 's. ". The article must be impersonal; the authors must avoid the use of personal pronouns such as We, or our. The authors must improve the presentation of figures, the use of print screens is not recommended in formal documents.
Reviewer 2 Report
Title: Enabling Artificial Intelligent Virtual Sensors in an IoT Environment
Even Though this paper gives us the idea to replace physical sensors with virtual sensors. I have some comments over your article.
-
You can remove the pronoun “we” throughout this article. Good research articles never have such a word.
-
Keep abbreviations for comment terminologies. For example Artificial neural network can be written as Artificial Neural Network (ANN). after that whenever called, just use ANN.
-
For equations 1 and 2, you may expand the terminologies. For example, what is wt0? Like that for equations 3 and 4 too.
-
Mathematical expressions of chosen ML algorithms are missing. At Least for RANN can we express it with mathematical expressions.
-
What is the reason for comparing almost 10 plus ML algorithms? Do not imagine the time consumption to predict the results.
Reviewer 3 Report
The topic of artificial intelligence is discussed in the manuscript in relation to the Internet of Things. Before the paper can be accepted, there are a few concerns that need to be addressed and resolved. The following components need to be included:
1- The abstract needs to be improved to include the problem as well as suggestions on how to solve it.
2- Four distinct layers make up the standard IoT architecture: the device layer, the network layer, the platform layer, and the service layer. I think a diagram would be helpful in illustrating the overall structure.
3- Related work is lacking and should be improved. Discuss other research gaps, and impacts.
4- in line 161-162, "There are three main types of approaches to design a virtual sensor model: mechanism-based, knowledge-based and data-driven, with the data-driven methods to have become the mainstream approach" More explanation is required, as well as what distinguishes each method.
5- In line 167, Brunello et al.'s work [18] is the closest work to our paper. More in-depth analysis is needed to illustrate the difference between the previous work and the proposed model. Also, I did not find the comparison between the previous work and the new proposed model.
6- Lines 179-182 require a comprehensive description of each item.
7- The manuscript contains some grammatical errors.
Round 2
Reviewer 3 Report
All of the reviewers' comments on the proposed model have been incorporated into the update manuscript.